# Monitoring progress towards elimination of hepatitis B and C in the EU/EEA

**Katherine C. Sharrock[1], Teymur Noori[2], Maria Axelsson[3], Maria Buti[4], Asuncion Diaz[5], Olga Fursa[6], Greet Hendrickx[7], Cary James[8], Irena Klavs[9], Marko Korenjak[10], Mojca Maticic[11,12], Antons Mozalevskis[13], Lars Peters[14], Rafaela Rigoni[15], Magdalena Rosinska[16], Kristi Ruutel[17], Eberhard Schatz[15], Thomas Seyler[18], Irene Veldhuijzen[19], Erika Duffell[2]***

**1** National AIDS Trust, London, United Kingdom, **2** European Centre for Disease Prevention and Control, Solna, Sweden, **3** Department of Public Health Analysis and Data Management, Public Health Agency of Sweden, Solna, Sweden, **4** Liver Unit, Valle d'Hebron University Hospital, Ciberehd del Insituto Carlos III Barcelona, Barcelona, Spain, **5** Instituto de Salud Carlos III, Madrid, Spain, **6** Centre of Excellence for Health Immunity and Infections Copenhagen, Copenhagen, Denmark, **7** Viral Hepatitis Prevention Board, Centre for the Evaluation of Vaccination, Vaccine & Infectious Diseases Institute, University of Antwerp, Antwerp, Belgium, **8** World Hepatitis Alliance, London, United Kingdom, **9** National Institute of Public Health, Ljubljana, Slovenia, **10** European Liver Patients' Association, Brussels, Belgium, **11** Faculty of Medicine, University of Ljubljana, Ljubljana, Slovenia, **12** Clinic for Infectious Diseases and Febrile Illnesses, University Medical Centre Ljubljana, Ljubljana, Slovenia, **13** World Health Organization, Regional Office for Europe, Copenhagen, Denmark, **14** CHIP, Centre of Excellence for Health, Immunity and Infections, Rigshospitalet, University of Copenhagen, Copenhagen, Denmark, **15** Correlation European Harm Reduction Network, Amsterdam, the Netherlands, **16** Department of Epidemiology of Infectious Diseases and Surveillance, National Institute of Public Health NIH – National Research Institute, Warsaw, Poland, **17** National Institute for Health Development, Tallinn, Estonia, **18** European Monitoring Centre for Drugs and Drug Addiction, Lisbon, Portugal, **19** National Institute for Public Health and the Environment (RIVM), Centre for Infectious Disease Control, Bilthoven, the Netherlands

* Erika.duffell@ecdc.europa.eu

**Data Availability Statement:** All data presented is available in an online report available at https://www.ecdc.europa.eu/en/publications-data/monitoring-responses-hepatitis-b-and-c-epidemics-eueea-countries-2020-data.

## Abstract

This paper presents data on selected indicators to show progress towards elimination goals and targets for hepatitis B and hepatitis C in the 31 countries of the European Union (EU) and European Economic Area (EEA). A monitoring system was developed by the European Centre for Disease Prevention and Control, which combined newly collected data from EU/EEA countries along with relevant data from existing sources. Data for 2017 were collected from the EU/EEA countries via an online survey. All countries provided responses. In 2017, most countries reporting data had not reached prevention targets for childhood hepatitis B vaccination and for harm reduction services targeting people who inject drugs (PWID). Four of 12 countries had met the target for proportion of people living with chronic HBV diagnosed and seven of 16 met this target for hepatitis C. Data on diagnosed cases treated were lacking for hepatitis B. Of 12 countries reporting treatment data for hepatitis B, only Iceland met the target. This first collection of data across the EU/EEA highlighted major issues with data completeness and quality and in the indicators that were used, which impairs a clear overview of progress towards the elimination of hepatitis. The available data, whilst incomplete, suggest that as of 2017, the majority of the EU/EEA countries were far from meeting most of the 2020 targets, in particular those relating to harm reduction and diagnosis. It is critical to improve the data collected in order to develop

**Funding:** This work was funded by the European Centre for Disease Prevention and Control (ECDC), Framework contract number ECDC/2019/037. The funders worked in close collaboration with the contractor to undertake the work and was involved in all aspects of the project.

**Competing interests:** The authors have declared that no competing interests exist.

more effective services for hepatitis prevention, diagnosis, and treatment that are needed in order to meet the 2030 elimination targets.

## Introduction

In 2015, the United Nations Member States adopted the Sustainable Development Goals (SDGs) for 2030 [1]. The SDGs are comprised of 17 goals and 179 targets, including goal 3, "to promote health and wellbeing", and target 3.3: "End the epidemics of AIDS, tuberculosis, malaria, and neglected tropical diseases and combat hepatitis, waterborne and other communicable diseases". Chronic viral hepatitis caused by hepatitis B virus (HBV) and hepatitis C virus (HCV) is an important cause of morbidity and mortality globally, including in Europe. It was estimated in 2017 that in the European Union (EU) (including the United Kingdom (UK)) and European Economic Area (EEA), approximately 4.7 million people were living with chronic HBV infection, 3.9 million people with chronic HCV infection [2], and deaths attributed to HBV and HCV accounted for 55% of liver cancer deaths and 45% of cirrhosis and other chronic liver disease deaths [3].

In 2016, the 69th World Health Assembly endorsed the first Global Health Sector Strategy (GHSS) for viral hepatitis, with the goal of eliminating viral hepatitis as a major threat to public health by 2030, in alignment with the SDGs [4]. Elimination is defined as a 65% reduction in hepatitis-related deaths and a 90% reduction in new chronic HBV and HCV infections compared to the 2015 baseline. The GHSS lays out global targets for services (five prevention measures, testing, and treatment) and impact (incidence and mortality) for 2020 and 2030 to guide elimination efforts.

The first action plan for the health sector response to viral hepatitis in the World Health Organization (WHO) European Region, published in 2017, adapts the GHSS for the context of the countries of the WHO European Region, taking epidemiological, political, and social factors into account and identifying a strategic framework [5]. The European action plan also specifies service and impact interim targets for 2020, some of which are more ambitious than the global targets (e.g. in relation to the proportion who should be diagnosed) in recognition of the existing prevention and control efforts in the WHO European Region and the capacity of existing systems to further impact on progress against the epidemics (see Table 1).

In 2017, the European Centre for Disease Prevention and Control (ECDC), in collaboration with WHO Regional Office for Europe, the European Hepatitis Network and other key stakeholders, established a system to monitor progress towards global and regional targets for the elimination of hepatitis B and hepatitis C in the EU/EEA. Prior to this system being established there was no system across the EU/EEA countries to collate and collect data to assess progress towards elimination for hepatitis. We defined a small set of "core indicators" essential to monitoring progress towards the elimination of viral hepatitis in line with SDG target 3.3.

The objectives of this paper are to 1) describe the available data on the core indicators for monitoring progress towards the elimination of HBV and HCV in the EU/EEA; 2) identify gaps in the data and ways to improve monitoring in the future; and 3) interpret the findings to identify priority areas for action to achieve the goal of elimination.

## Materials and methods

The hepatitis B and C monitoring system developed by ECDC for EU/EEA countries combined data from existing sources, such as WHO vaccination coverage data, with newly

**Table 1. Core indicators for measuring progress towards the SDG targets and related 2020 targets.**

| Indicator | WHO European Region Action Plan Target for 2020 | WHO GHSS Target for 2020 |
|---|---|---|
| Prevention | | |
| HBV: Coverage with three doses of the HBV vaccine among 1 year olds in countries that implement universal childhood HBV vaccination | 95% vaccinated | 90% vaccinated |
| HCV: Harm reduction for people who inject drugs (PWID) <br>• Coverage of clean needle and syringe programmes (NSP) <br>• Coverage of opioid substitution therapy (OST) | • 200 syringes distributed per PWID per year; <br>• 40% of opioid dependent PWID receiving opioid substitution therapy | 200 sterile needles and syringes provided per PWID per year |
| The continuum of care for HBV and HCV | | |
| Percent of those with chronic infection tested and aware of their diagnosis | 50% diagnosed* | 30% diagnosed |
| Percent of those aware of their HBV diagnosis on treatment, among those eligible for treatment** Percent of those aware of their HCV diagnosis started on treatment | 75% on treatment/started on treatment*** | 5 million people receiving HBV treatment 3 million people have received HCV treatment |
| Percent of those on treatment achieving viral suppression (HBV) or of those on treatment achieving sustained viral response (HCV) | 90% achieve viral suppression (HBV) or sustained viral response (HCV) | NA |

HBV- hepatitis B virus; HCV–hepatitis C virus; PWID–people who injects drugs; SDG–sustainable development goals.

*Diagnosed with chronic hepatitis (HBV: HBsAg positive; HCV: PCR positive).

** According to European Association for the Study of the Liver (EASL) guidelines.

***Started on treatment during the reporting year.

NB: WHO target around OST does not further define measurement of OST coverage.

collected data. The system aims to present a comprehensive description of the hepatitis B and C epidemics, and the responses to these epidemics, while minimising the reporting burden on countries. ECDC, in consultation with an advisory group, identified and defined indicators that were aligned with the milestones and targets of the WHO European Regional action plan [5]. Wherever possible, indicators were harmonised with the global indicators developed by WHO [6].

Existing data sources were mapped against the overall set of indicators identified by the advisory group and this mapping identified relevant data from several EU/EEA agencies and projects and other data sources that could be collated for reporting (S1 Table). The most recent data were collated from each of these data sources, which included: ECDC (epidemiological data and data on vaccination and infection prevention and control programmes); the European Monitoring Centre for Drugs and Drug Addiction (EMCDDA) (data on harm reduction and the PWID population) and WHO/UNICEF Joint Reporting Process (vaccination coverage data). A detailed overview of these sources is found in the *Supplementary Materials* section (S1 Table) and a full description of the data included from these sources is detailed in the ECDC report [7].

A survey was developed to collect country-level data on indicators for which there was no data already available from other agencies, which mostly consisted of data related to testing and treatment. The online survey was built using EUSurvey software, pilot tested, and sent by ECDC to the formally nominated ECDC hepatitis focal points in the 31 countries comprising the EU/EEA (at the time also including the UK). These focal points are based in the national

public health institute or the ministry of health. Data collection from nominated expert country focal points and data cleaning and validation occurred between December 2018 and August 2019. The focal points were asked to provide data for 2017, or the most recent available data, and for the data provided to be at the national level. Ethics review and approval was not required to carry out this survey.

A subset of core indicators was defined (Table 1) for monitoring progress towards hepatitis elimination in line with the SDGs from among all the indicators collected in the monitoring system. These indicators include, for HBV and HCV each, one primary prevention indicator and the continuum of care indicators relating to diagnosis, treatment, and viral suppression (HBV)/sustained virologic response (SVR) (HCV). These indicators pertain to programmatic service targets. Incidence and mortality were not included as core indicators because data were not available directly from countries at the time. Data analysis consists of presenting completeness of data reporting and progress towards the 2020 targets for each of the core indicators.

## Results

### Completeness and data sources

For data collected directly from countries, the overall response rate was 100%, with all 31 EU/EEA countries providing a response. Tables 2 and 3 include a summary of the data provided by countries to EMCDDA, WHO and ECDC on the WHO European Region Action plan targets for the core indicators. The indicators with the best data availability included routine childhood hepatitis B vaccination coverage, with 24 of the 27 countries that had a universal childhood vaccination programme in 2017 reporting data, and coverage of opioid substitution therapy (OST), with 18 of the 29 countries that report to EMCDDA reporting. Data were most lacking in the continuum of care indicators and especially the HBV continuum of care, with only three of 31 countries reporting the proportion of those treated with viral suppression.

Although data was requested for 2017, or the most recent year, from the information provided estimates of the numbers with chronic hepatitis B and C were over five years old in around a third of countries (5/18 hepatitis B; 7/23 hepatitis C). In relation to the data on treatment, one country provided data prior to 2017 for hepatitis B treatment and three countries provided data prior to 2017 for hepatitis C.

Information were obtained on the source of the data provided, but was lacking for many of the data points [7]. The source of data reported for the estimates of cases with chronic hepatitis B and C were mostly prevalence surveys but a few countries cited that the data arose from mathematical modelling (e.g. multi-parameter evidence synthesis methods) or from routine case based surveillance systems. For the diagnosis data, most countries for both infections reported that the source of data came from surveillance data, although a few countries indicated that the source was a survey or the data had been derived through modelling. The sources of treatment data varied and included data from national health insurance sources, clinical databases, cohort studies and drug databases.

### Policy

Countries were asked directly whether a national plan or strategy existed that covered the response to viral hepatitis. Of the 31 countries, 20 (65%) reported there was a plan and 9 of 18 (50%) countries with information on funding reported that there were funds allocated from the national budget to implement the plan (Croatia, France, Iceland, Latvia, Luxembourg, Portugal, Slovakia, Spain).

**Table 2. Region and country reporting and progress towards targets for prevention of new hepatitis B and C infections in the EU/EEA and UK, 2017¥ [8]+.**

| Country | Universal childhood hepatitis B vaccination | Prevention targets | | |
|---|---|---|---|---|
| | | Hepatitis B vaccine coverage of three doses in children (%) | # syringes distributed /PWID | OST coverage (%) |
| *TARGET* | | *95%* | *200* | *40%* |
| Source of data | | WHO [9] | EMCDDA [10] | |
| Countries reporting data | | 24/27 | 15/29 | 18/29 |
| Countries meeting target | | 7 | 4 | 11 |
| Austria | Yes | 90 | No data | **50** |
| Belgium | Yes | **97** | 50 | No data |
| Bulgaria | Yes (newborn) | 92 | No data | No data |
| Croatia | Yes | 94 | 192 | **54** |
| Cyprus | Yes | **97** | 1 | 18 |
| Czechia | Yes | 94 | 147 | 38 |
| Denmark | No | NA* | No data | No data |
| Estonia | Yes | 92 | **232** | No data |
| Finland | No | NA* | **373** | No data |
| France | Yes | 90 | 102 | **85** |
| Germany | Yes | 87 | No data | **54** |
| Greece | Yes | **96** | 76 | **65** |
| Hungary | Yes (adolescent) | No data | 21 | No data |
| Iceland | No | NA* | NA** | NA** |
| Ireland | Yes | **95** | No data | **54** |
| Italy | Yes | 94 | No data | 30 |
| Latvia | Yes | **98** | 108 | 9 |
| Liechtenstein | Yes | NA** | NA** | NA** |
| Lithuania | Yes (newborn) | 94 | 28 | 15 |
| Luxembourg | Yes | 94 | **305** | **66** |
| Malta | Yes | 88 | No data | **72** |
| Netherlands | Yes | 92 | No data | No data |
| Norway | Yes | No data | **332** | No data |
| Poland | Yes (newborn) | 93 | No data | 18 |
| Portugal | Yes (newborn) | **98** | 108 | **45** |
| Romania | Yes (newborn) | 92 | No data | 8 |
| Slovakia | Yes | **96** | No data | No data |
| Slovenia | Yes | 89 | No data | **62** |
| Spain | Yes | 93 | 119 | No data |
| Sweden | Yes | 76*** | No data | No data |
| United Kingdom | Yes | No data | No data | **57** |

¥ Data provided for 2017 or most recent year.

+ Bold text indicates the target was met.

* Not applicable because country did not have a universal childhood HBV vaccination programme.

** Not applicable because country was not included in data collection efforts (data reported through to Switzerland).

***Sweden only started national vaccination programme in 2017.

Source: Monitoring the responses to hepatitis B and C epidemics in the EU/EEA Member States, 2019 [8]. WHO [9].

NA–Not applicable; EMCDDA–European Monitoring Centre for Drugs and Drug Addiction.

**Table 3. Region and country reporting and progress towards targets for the hepatitis continuum of care in the EU/EEA and UK, 2017¥ [8]⁺.**

| Country | Hepatitis B Continuum of care | | | Hepatitis C Continuum of care | | |
|---|---|---|---|---|---|---|
| | % diagnosed | % of diagnosed on treatment | % with VS | % diagnosed | % of diagnosed started on treatment | % with SVR |
| *TARGET* | *50%* | *75%* | *90%* | *50%* | *75%* | *90%* |
| Source of data | ECDC [6] | | | ECDC [6] | | |
| Countries reporting data | 12/31 | 6/31 | 3/31 | 16/31 | 12/31 | 12/31 |
| Countries meeting target | 4 | NA⁺⁺ | 1 | 7 | 1 | 12 |
| Austria | No data | 61 | No data | 35 | 28 | No data |
| Belgium | No data | No data | No data | No data | No data | No data |
| Bulgaria | 2 | **100** | 29 | 4 | 36 | **100** |
| Croatia | No data | No data | No data | 25 | 6 | **99** |
| Cyprus | No data | No data | No data | No data | No data | No data |
| Czechia | No data | No data | No data | No data | No data | No data |
| Denmark | **72** | No data | No data | 44 | No data | No data |
| Estonia | 11 | No data | No data | 29 | No data | No data |
| Finland | No data | No data | No data | No data | No data | No data |
| France | 18 | No data | No data | **81** | 18 | **90** |
| Germany | No data | No data | No data | No data | No data | No data |
| Greece | 35 | No data | No data | 20 | No data | No data |
| Hungary | No data | No data | No data | 26 | 8 | **93** |
| Iceland | No data | No data | No data | **77** | **100** | **95** |
| Ireland | **57** | No data | No data | **63** | 10 | **98** |
| Italy | No data | No data | No data | No data | No data | No data |
| Latvia | 24 | No data | No data | **97** | 6 | **91** |
| Liechtenstein | No data | No data | No data | No data | No data | No data |
| Lithuania | No data | No data | No data | No data | No data | No data |
| Luxembourg | No data | No data | No data | No data | No data | No data |
| Malta | No data | No data | No data | No data | No data | No data |
| Netherlands | **52** | 22 | No data | No data | No data | No data |
| Norway | No data | 4 | No data | **71** | No data | No data |
| Poland | 47 | No data | No data | 25 | No data | No data |
| Portugal | No data | No data | No data | No data | No data | **97** |
| Romania | 7 | 22 | 86 | 8 | 42 | **100** |
| Slovakia | 15 | No data | No data | **93** | No data | No data |
| Slovenia | No data | 28 | **100** | No data | 14 | **95** |
| Spain | No data | No data | No data | No data | 36 | **100** |
| Sweden | No data | No data | No data | No data | No data | No data |
| United Kingdom | 59 | No data | No data | **54** | 16 | **96** |

¥ Data provided for 2017 or most recent year. If no year provided, data is from 2017. Year provided for all other data.

⁺ Bold text indicates the target was met.

⁺⁺ Estimate does not discount cases not eligible for treatment according to clinical guidelines.

Source: Monitoring the responses to hepatitis B and C epidemics in the EU/EEA Member States, 2019 [8].

NA–Not applicable; ECDC–European Centre for Disease Prevention and Control.

## Prevention

**i. Hepatitis B vaccine coverage.** In 2017, 27 countries in the EU/EEA recommended universal childhood vaccination against hepatitis B as part of their national vaccination programme [11]. In 2017, Denmark, Finland, and Iceland did not have a national policy for

universal vaccination and Sweden had subnational policies for universal hepatitis B vaccination (Sweden now has a national policy). Of the 27 countries with a universal childhood HBV vaccination programme, Hungary, Malta, and Slovenia offered the vaccine outside of the primary infant schedule in regard to the timing of doses [11].

Of 24 countries reporting vaccine coverage in 2017 [10], seven countries had reached the 2020 target of 95% childhood vaccine coverage, although many additional countries had vaccine coverage approaching 95% (Fig 1).

**ii. Harm reduction for people who inject drugs.** Fifteen countries (52%) reported on estimated number of clean syringes provided per person who injects drugs (PWID) in 2017 among 29 countries surveyed by EMCDDA [12]. Four out of the 15 countries reporting data for 2017 had coverage of at least 200 syringes distributed per PWID per year (Fig 2). Eighteen countries out of 29 (62%) had estimates of the population of high-risk opioid users and coverage of OST within this population for 2017. Eleven countries reported that the 2020 target had been met, with at least 40% coverage of OST, including Austria, Croatia, France, German, Greece, Ireland, Luxembourg, Malta, Portugal, Slovenia, and the UK (Fig 3).

## Continuum of care

**i. Chronic HBV and HCV cases diagnosed.** The proportion of chronic HBV and HCV cases that were diagnosed as of the end of 2017 was calculated as a fraction with the number diagnosed as the numerator and the estimated number with chronic infection as the denominator for each country reporting these data in the ECDC survey.

Twelve countries of 31 (39%) had data for both the numerator and denominator for HBV and as of 2017, four (Denmark, Ireland, Netherlands, UK (Scotland)) of the 12 countries had met or exceeded the 50% target for proportion of people living with chronic HBV diagnosed (Fig 4).

Sixteen countries (52%) reported data on both the estimated number of people living with chronic HCV infection and the number diagnosed via RNA test. Seven of the 16 countries (France, Iceland, Ireland, Latvia, Norway, Slovakia, UK (England and Scotland)) had met the 50% target in 2017 (Fig 5).

**ii. Eligible cases treated.** Six out of 31 countries (19%) provided data on both the number of people with chronic HBV infection diagnosed to the end of 2017 and the number on treatment in 2017 (Austria, Bulgaria, the Netherlands, Norway, Romania, and Slovenia). However, many cases diagnosed with chronic HBV infection are not eligible for treatment according to clinical guidelines. Romania was the only country that provided an estimate of the total number of HBV cases that were eligible for treatment, although did not specify how many of these cases eligible for treatment had been diagnosed. So overall, no country reported data on both the diagnosed HBV cases eligible for treatment and the number of these receiving.

Twelve of 31 countries (39%) reported data on both the number of people with diagnosed chronic HCV infection to the end of 2017 and the number of those diagnosed who were started on antiviral treatment in the year 2017. Among the 12 countries reporting, only Iceland met the target of having 75% of those diagnosed with chronic HCV on treatment (Fig 6).

**iii. Sustained virologic response/viral suppression among patients treated.** For chronic HBV cases, only three countries (Bulgaria, Romania, Slovenia) reported data on the proportion of those treated who attained viral suppression. Slovenia was the only country to reach the 2020 target of having 90% of treated patients with viral suppression in 2017.

Twelve of 31 countries (39%) reported data on numbers with chronic HCV treated and numbers who had a sustained virologic response. All 12 countries had reached the 2020 target of having 90% of treated patients with sustained virologic response in 2017. These countries

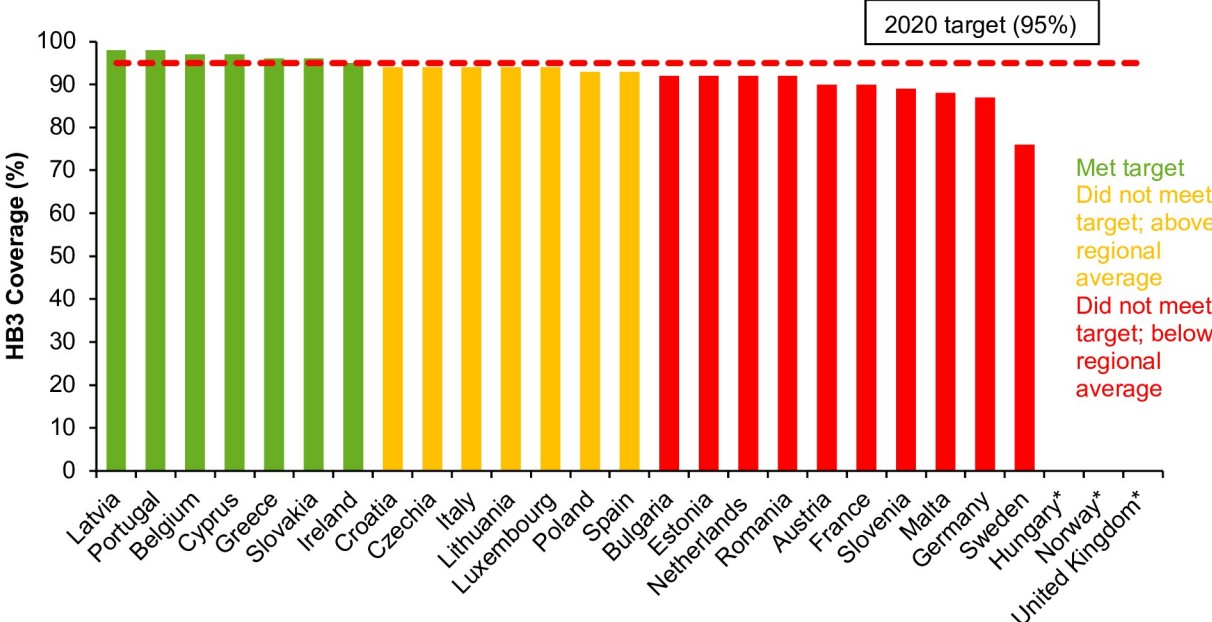

**Fig 1. Coverage (%) of three doses of HBV vaccine (HB3) in EU/EEA countries that implement universal HBV vaccination in 2017\*#+ [7].**
\*No data available from Hungary, Norway, and the United Kingdom. No data available from Hungary as the programme is a two-dose regime provided from the age of 13 years. ♯ Data for Austria based on HB3 coverage among children aged four years. + National programme in Sweden only implemented during 2016 and in the United Kingdom from 2017 (with estimated coverage in 2019 >90%).

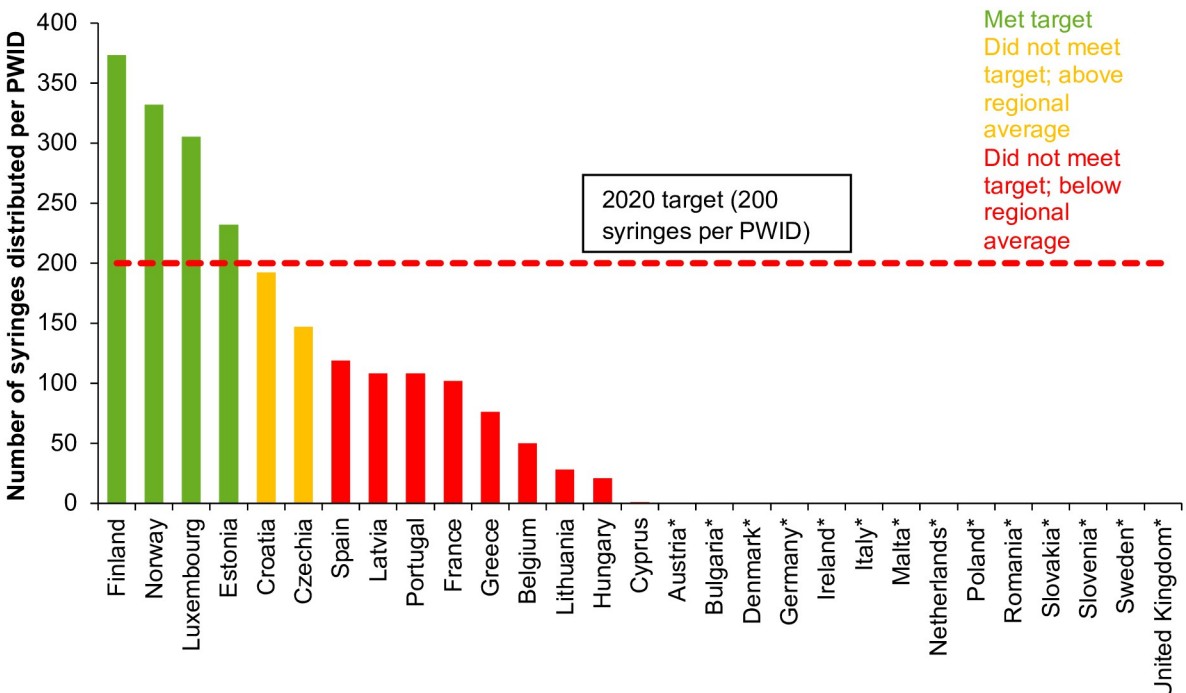

**Fig 2. Estimated number of syringes provided per person who injects drugs in 2017 or most recent year with available data [11]\*\*.** \*No data from Austria, Bulgaria, Denmark, Germany, Ireland, Italy, Malta, the Netherlands, Poland, Romania, Slovakia, Slovenia, Sweden and the United Kingdom. \*\*Iceland and Lichtenstein were not surveyed in this source of data.

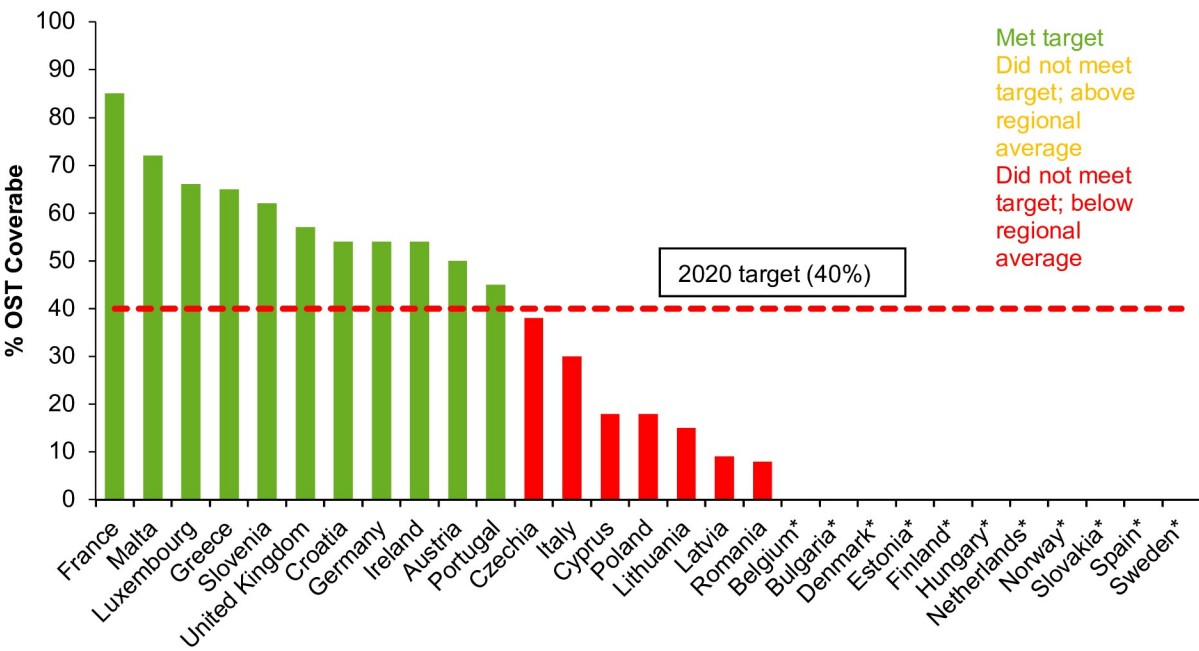

**Fig 3. Percentage of estimated high-risk opioid users receiving opioid substitution treatment in 2017 or most recent year with available data** [11]**. *No data from Belgium, Bulgaria, Denmark, Estonia, Finland, Hungary, the Netherlands, Norway, Slovakia, Spain and Sweden. **Iceland and Lichtenstein were not surveyed in this source of data.

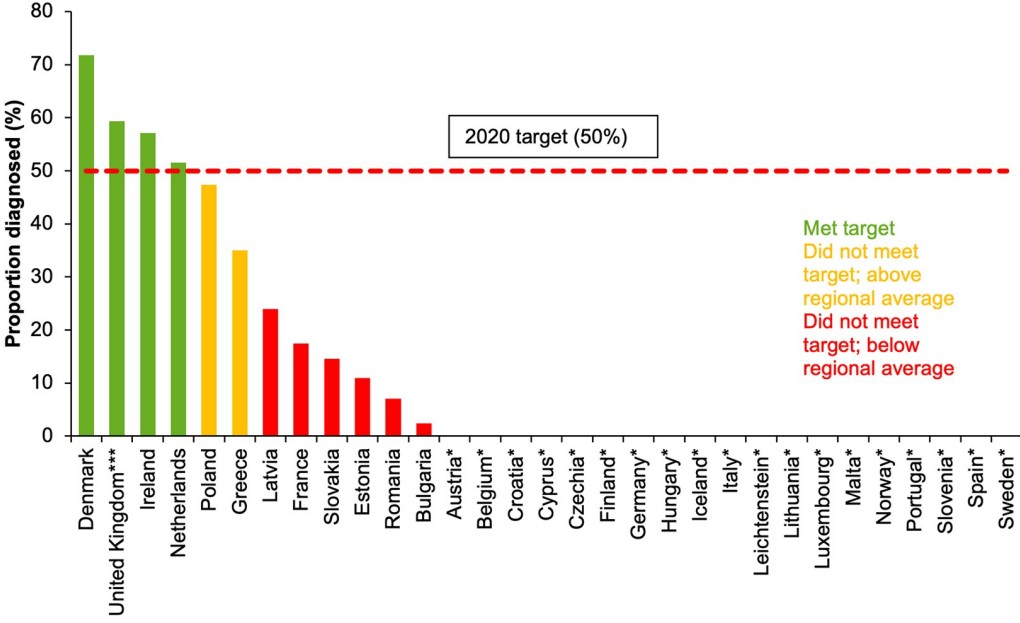

**Fig 4. Proportion (%) of people living with chronic HBV infection who had been diagnosed in EU/EEA countries**, 2017* [8].** * No data from Austria, Belgium, Croatia, Cyprus, Czech Republic, Finland, Germany, Hungary, Iceland, Italy, Liechtenstein, Lithuania, Luxembourg, Malta, Norway, Portugal, Slovenia, Spain, Sweden. **Data incomplete on diagnosed cases from Bulgaria (data from 2016), Estonia (data from 2004) and the Netherlands (data from 2000). ***Data represent Scotland only.

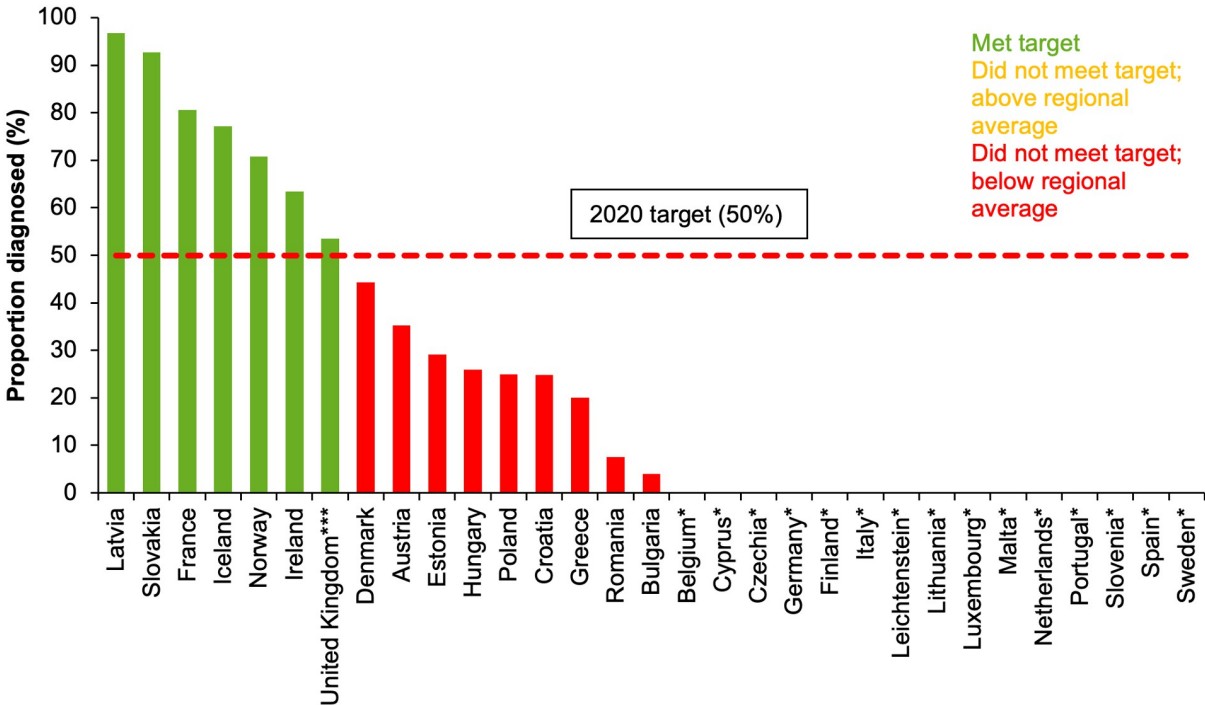

**Fig 5. Proportion (%) of people living with chronic HCV infection who had been diagnosed in EU/EEA countries**, 2017* [8].** * No data from Belgium, Cyprus, Czechia, Germany, Finland, Italy, Liechtenstein, Lithuania, Luxembourg, Malta, the Netherlands, Portugal, Slovenia, Spain, Sweden. **Data incomplete on diagnosed cases from Austria (data from 2009), Bulgaria (data from 2016), Estonia (data from 2004) and Spain (data from 2015). For some countries data include cured/spontaneously resolved cases. ***Data represent England and Scotland.

were Bulgaria, Croatia, France, Hungary, Iceland, Ireland, Latvia, Portugal, Romania, Slovenia, Spain, and the UK (England).

## Discussion

The data presented here are a subset of the indicators collected and collated in the first round of a new monitoring system for hepatitis B and C developed for EU/EEA countries. The monitoring system was successful in minimising reporting burden utilising existing data where available, but it clearly demonstrated areas where information is lacking and where greater collaboration is needed between clinicians and public health professionals to improve the quality and availability of the data.

The results show that efforts to prevent the transmission of HBV and HCV must be improved and sustained to reach elimination targets. Universal vaccination of children against HBV has been recommended by WHO for many years but data on vaccination coverage showed great variation, with only seven countries had achieved the 2020 target in 2017, although many of the other countries had vaccine coverage approaching the target. Future coverage of vaccination for HBV may be affected by vaccine hesitancy and the "anti-vaccination" movement in Europe seen during the recent COVID-19 pandemic, together with the direct impact of the pandemic on the delivery of routine vaccinations [13,14]. The success of any future efforts aimed at increasing vaccination uptake will require effective engagement with the community to address these barriers [8]. Vertical transmission is now an uncommon route of HBV transmission within the EU/EEA [15], but high coverage of birth dose vaccination remains important to minimise potential transmission in countries that implement universal

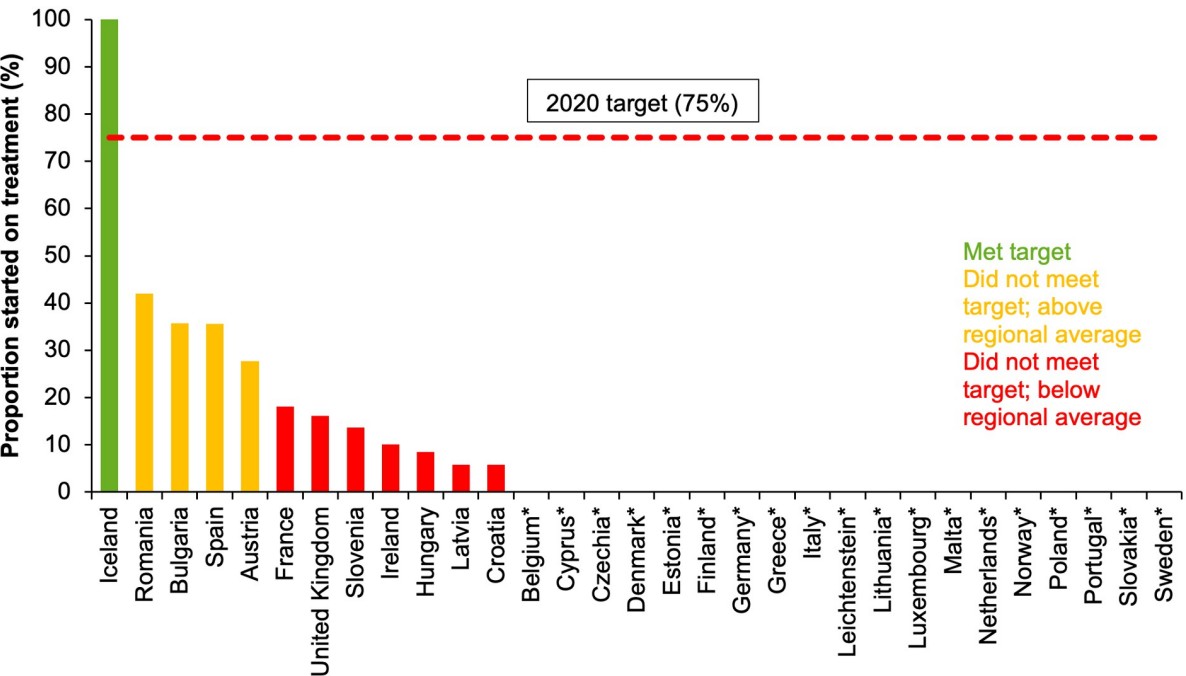

**Fig 6. Proportion of people diagnosed with chronic HCV infection who have been started on treatment in 2017 in EU/EEA countries and the UK**[*] **[8].** [*]No data from Belgium, Cyprus, Czechia, Denmark, Estonia, Finland, Germany, Greece, Italy, Liechtenstein, Lithuania, Luxembourg, Malta, Netherlands, Norway, Poland, Portugal, Slovakia, Sweden.

birth dose vaccination as well as in those countries that adopt a targeted approach focused on babies born to mothers with chronic infection. Although the data relating to the prevention of vertical HBV transmission are not detailed here, these data were collected by WHO Europe and incorporated into our monitoring system with the data showing that four of the five countries in the region (Bulgaria, Lithuania, Poland, Portugal, and Romania) that implement universal new-born vaccination had reached the 2020 target of 90% coverage [7]. Among the 26 EU/EEA countries with a targeted approach, antenatal screening coverage data were only available from five countries, with four of these countries having already reached the 2020 target of 90% screening [7]. The lack of data relating to antenatal screening, including the follow up of babies born to positive mothers, hampers a clear picture of the effectiveness of these programmes.

Harm reduction indicators were chosen as the primary prevention indicators for HCV because the injection of drugs accounts for a large proportion of HCV transmission in Europe [16] and is a transmission route for HBV as well [15]. There remains a need to scale up harm reduction services for the prevention of HBV and HCV among PWID, as there was evidence of suboptimal implementation among the countries reporting 2017 data. This is likely due to political and legal environments that criminalise PWID, stigma and discrimination against PWID, and insufficient funding for harm reduction programmes, despite gradual progress towards an improved harm reduction policy environment and good evidence of the cost-saving nature of these interventions [17–19].

Timely antiviral treatment of PWID infected with HCV is also critical for preventing HCV transmission and achieving elimination. Data on treatment of PWID will be collected in the next data collection in 2023 to provide an overview of progress, but whilst restrictions for HCV treatment based on fibrosis have now been lifted, it remains of concern that recently

published information indicates existing restrictions in seven EU countries [20] for those actively using drugs or not enrolled in OST [21]. Of additional concern is the negative impact of the COVID-19 pandemic on the provision of harm reduction and infectious disease testing and treatment services for PWID [22,23]. Although the COVID-19 pandemic has presented a major challenge to services, some countries have seized the opportunity to implement innovative strategies to provide services to PWID during the pandemic [24]. Some of these strategies have included outreach to bring services to PWID in the community, increased accessibility of care via telemedicine, increased length of prescriptions and take-home doses for OST [21,22], and hepatitis and human immunodeficiency virus (HIV) testing campaigns in temporary accommodations for homeless populations [25].

Despite the limitations of the data, countries in the EU/EEA appear to be far from meeting WHO 2020 diagnosis target of 50% with most people living with chronic HBV and HCV undiagnosed. It is critical that this gap in diagnosis be addressed, as diagnosis is the entry point into care and treatment. Testing needs to be scaled up, with a particular focus on testing populations at greatest risk of hepatitis infection including PWID, people in prisons, migrants, men who have sex with men, and people living with HIV in relevant settings, using appropriate testing technologies and methods, and guided by key performance indicators and local epidemiological and testing data. The screening of key populations with high prevalence of infection has been found to be cost effective [26–28]. However, testing services remain suboptimal in community-based settings and innovative strategies, like outreach service testing and peer-supported and -led services, could help overcome barriers and making testing more accessible [26–28]. In scaling up of screening of people in prison settings, active case finding and peer-supported interventions have been found to be effective [29,30]. Hepatitis testing using point of care testing in a variety of different healthcare settings may help to meet the varying needs of diverse populations [31,32]. There is increasing recognition of the importance of integrating HBV and HCV testing with HIV testing as well, given common modes of transmission, overlapping risk groups, and high levels of co-infection [33].

Currently, there is a lack of robust estimates of both the estimated numbers of people living with chronic HBV and HCV infection and the number of people living with diagnosed chronic HBV and HCV infection. Some estimates of numbers with current chronic infection are based on outdated data or weak sources, such as studies with poor sampling methodologies or blood donation data. Blood donors are generally not considered to be representative of the general population, due to self-selection and strict regulation by blood banks [2]. Studies to estimate prevalence in the general population and key risk groups should be conducted using a rigorous methodological design and for HCV these studies should determine active infection in addition to antibodies. Prevalence studies are often challenging to conduct well and often yield biased results due to the many difficulties in the methodological approach, such as sampling for example. To help address these challenges, ECDC has recently developed the SPHERE-C study protocol to support countries undertaking HCV prevalence studies in the general population [34] and is working with countries to conduct surveys using different methodological approaches including the piggybacking of hepatitis testing onto planned COVID-19 serosurveys. National estimates of prevalence based on prevalence estimates among different groups that also take into account any changes to the prevalence pool from death, migration, and treatment, are needed and ECDC is currently working to produce these estimates [34]. Modelling studies are also key to help fill data gaps in countries that lack empirical data. For HCV, the proportion diagnosed is challenging to interpret as there are changes to the denominator (number living with chronic HCV) over time as individuals are treated and cured. The indicator collected in the survey also did not include diagnosis by testing for HCV core antigen that is recommended as an affordable alternative to HCV RNA testing [35]. Additionally, in some

of the reporting countries, data on numbers diagnosed were only available for the previous few years, resulting in an under-estimation of the numerator.

For HBV, no country was able to report data on the proportion on treatment among those diagnosed and eligible for treatment, so for HBV it is not possible to measure progress towards the 2020 treatment target of having 75% on treatment. According to current international guidelines, not all those with chronic HBV infection need to receive treatment; eligibility for treatment depends on serum HBV DNA and alanine aminotransferase (ALT) levels and severity of liver disease [36]. Those who do not meet indications for treatment should still be clinically monitored and retained in care. These characteristics of the epidemiology of chronic HBV infection differentiate the HBV continuum of care from that of HCV or HIV, for which all those living with chronic infection benefit from treatment and qualify for treatment according to regional guidelines [37]. Future data collection efforts across the region would benefit from the inclusion of data on numbers with chronic HBV eligible for treatment, numbers retained in care among those not eligible, and numbers receiving treatment among those eligible. For HCV, of the 12 countries reporting data for 2017, only Iceland had achieved the 2020 target of 75% of those diagnosed being treated. However, the data on diagnosis not excluding those with cleared infection would result in an under-estimation of the proportion of those receiving treatment among those diagnosed. The difficulty of adjusting the estimate accordingly is a major challenge in accurately estimating proportion diagnosed with chronic HCV infection started on treatment in 2017. In addition, the use of a treatment indicator that includes the numbers diagnosed as the denominator can benefit countries with a low diagnosis rate so alternative indicators should also be considered.

For HBV, data on the proportion of cases who attained viral suppression was incomplete, with only three countries reporting. This lack of data does not allow for an assessment of progress towards the HBV viral suppression target for the EU/EEA, although given the high effectiveness of current drugs it is likely that the majority of those on treatment are virally suppressed [35]. The best progress along the continuum of care has been seen in rates of SVR (cure) among people treated for chronic HCV infection, with 100% of the 12 countries reporting data meeting or exceeding the 2020 target of 90% cured. This is to be expected, given the high efficacy of direct acting antivirals (DAAs) that are now in wide use [35]. Given that these DAAs are now the standard for treatment of chronic HCV infection, it is perhaps less important to monitor proportion with sustained virologic response in the HCV continuum of care framework.

The large gaps in available data highlight a critical need to improve viral hepatitis information systems in many countries. For most of the subset of indicators chosen to monitor progress towards the SDG 2030 targets, there were large numbers of countries that were unable to report 2017 data. In addition, data for many of the indicators came from a variety of different sources that are likely to have varying degrees of quality that were unable to be assessed as part of this work but this is an issue that needs to be reviewed. The largest gaps in the data were in the continuum of care data, especially for HBV percent on treatment and percent with viral suppression. HCV treatment and cure may have better data availability than HBV due to the recent advancements in treatment with DAAs which has opened the possibility of elimination. Data availability would be improved by country investment in data collection for surveillance and monitoring with dedicated expert staff and input from multi-sector stakeholders.

The survey findings on the national level funding and policy environment show there is a need to improve the policy and funding landscape in most countries [7]. Another essential factor is whether a country has invested in data collection for surveillance and monitoring with dedicated expert staff and input from multi-sector stakeholders. In Scotland, United Kingdom, strong advocacy led to the building of political consensus to support evidence-based policy

initiatives for addressing the HCV epidemic [38]. A case study on Scotland's efforts highlights a number of elements that made Scotland's HCV response successful, including a project management approach, a robust monitoring and surveillance system to inform actions and measure impact, adequate funding for a phased national strategy, and a multi-agency approach focused on addressing inequalities [38].

The monitoring system has several limitations that make it difficult to compare countries and to draw conclusions on progress towards targets. Firstly, data comparability is an important issue and while the questionnaire used standardised indicator definitions, some of the data submitted by countries to the monitoring survey came from disparate sources and timepoints (see report [7] for details). The provision of outdated data is a critical issue for hepatitis C on account of the need to adjust estimates of numbers infected and diagnosed as cases are treated and cured. Secondly, the country focal points who coordinated data submission were the ECDC focal points based in national public health institutes or ministries of health and were mostly epidemiologists and not always well connected to clinical bodies who hold the data which often led to the gaps in data. The EMCDDA report, the source of the harm reduction indicators, highlights many of the same limitations [12]. In addition, we are aware of different data sources in some countries with different estimates, which highlights the need for local collaboration between public health and clinical groups to reach consensus around which sources should be used. Differences in country capacity and data collection systems are a major challenge and targeted resources are required to realize high quality data for hepatitis, as happened for HIV through the establishment of systems to monitor progress towards regional and global targets [39].

Data collection efforts carried out in 2021 will seek to improve the monitoring system through close collaboration between ECDC, EMCDDA, WHO and the EU/EEA Member States. Monitoring efforts will benefit from strengthening of these relationships and collaborations between the country focal points and different stakeholders and sectors with each country. ECDC is working in collaboration with clinical and public health teams in countries to develop a sentinel system based in clinical sites to improve data quality and ensure greater consistency. Even greater involvement from civil society organisations is needed to help improve quality and reporting of data and to provide contextual information (e.g. on local treatment policies) to enable the interpretation of the data. The 2021 data collection survey features several continuum of care indicators which have been updated and refined to address some of the limitations highlighted above.

In conclusion, the hepatitis B and C monitoring system set up for EU/EEA countries aimed to collect standardised data to provide assessment of progress towards the WHO and SDG hepatitis targets and to highlight areas for future action. We found major gaps in the available data for most indicators related to prevention and the continuum of care, impairing the ability to monitor progress towards targets. Whilst the sources of data for many of the indicators were varied and many countries were unable to provide recent data, this collection of data represents an important first step towards understanding the responses to hepatitis in the region. The data gaps need to be addressed as a priority through greater political commitment to improve country monitoring systems, ensuring adequate resources for public health information as well as greater collaboration between clinical and public health bodies. From the data that were available, there is evidence of sub-optimal implementation of universal childhood HBV vaccination programmes and harm reduction programmes for PWID. There is also evidence to suggest that a high proportion of people living with chronic hepatitis infections remain undiagnosed and, among those diagnosed, many have not been linked for treatment. The new guidance from WHO with the inclusion of absolute targets should ease assessment of progress towards elimination [40]. Nevertheless, monitoring progress is challenging and the

gaps we find in the data and responses to hepatitis across the EU/EEA, a region of high and middle income countries, raises concern about progress towards elimination in other areas globally where resources are more limited. For the EU/EEA, it remains clear that greater efforts are needed to meet the 2030 WHO targets in support of the SDG of combatting viral hepatitis.

## Supporting information

**S1 Table. Details of the existing data sources included in the monitoring system [6].** (DOCX)

## Acknowledgments

The authors would like to thank the following individuals for their efforts reporting the data used in the manuscript: Austria: Bernhard Benka, Irene Kaszoni-Rueckerl, Robert Sauermann, Daniela Schmid. Belgium: Sophie Quoilin. Bulgaria: Marieta Simonova, Mariya Tyufekchieva, Tonka Varleva. Croatia: Tatjana Nemeth Blažić, Adriana Vince. Cyprus: Petros Katsioloudes. Czech Republic: Jitka Castkova, Vratislav Nemecek. Denmark: Susan Cowan. Estonia: Kristi Rüütel, Jevgenia Epštein. Finland: Henrikki Brummer-Korvenkontio, Tuija Leino, Kirsi Liitsola. France: Cecile Brouard, Mathias Bruyand, Stella Laporal. Lionel Lavin. Germany: Sandra Dudareva, Thomas Harder, Christian Kollan, Katrin Kremer, Daniel Schmidt, Ida Sperle-Heupel, Gyde Steffen, Ruth Zimmermann. Greece: Georgia Nikolopoulou, George Papatheodoridis. Hungary: Mihály Makara, Zsuzsanna Molnár. Iceland; Thorolfur Gudnason, Sigurdur Olafsson. Ireland; Linda Cosgrove, Kevin Kelleher, Mary Kelleher, Joanne Moran, Niamh Murphy, Karen Logan, Lois O'Connor, Aisling O'Leary. Italy: Maria Elena Tosti, Francesco Paolo Maraglino, Barbara Suligoi. Latvia: Raina Nikiforova, Ieva Melišus. Liechtenstein: Andrea Leibold. Lithuania: Irma Caplinskiene, Ligita Jančorienė. Luxembourg: Vic Arendt, Carole Devaux, Laurence Guillorit, Patrick Hoffman. Malta: Jackie Maistre Melillo, Tanya Melillo. Netherlands: Irene Veldhuijzen. Norway: Hans Blystad, Hilde Kløvstad, Øivind Nilsen. Poland: Magdalena Rosińska, Małgorzata Stępień. Portugal: Isabel Aldir, Ana Cláudia Cordeiro Fernandes, Rui Tato Marinho. Romania: Adriana Pistol, Corina Silvia Pop, Odette Popovici. Slovakia: Mária Avdičová, Ľubomír Skladaný. Slovenia: Kastelic Andrej, Mitja Ćosić, Mojca Gobec, Irena Klavs, Janja. Krizman-Miklavcic, Zdenka Kastelic, Sandra Kosmac,; Tanja Kustec, Ines Kvaternik, Eva Leban, Snezna Levicnik, Mojca Matičič, Mario Poljak, Maja Socan, Marta Vitek. Spain: Raquel Boix Martínez, Asunción Díaz Franco. Sweden: Maria Axelsson, Josefine Lundberg Ederth. United Kingdom: England–Koye Balogun, Ian Hayden, Ross Harris, Helen Harris, Goergina Ireland, Sema Mandal; Scotland–David Goldberg, Sharon Hutchinson, Andrew McAuley, Allan McLeod, Shanley Smith, Lesley Wallace, Alan Yeung; Wales–Noel Craine, Brendan Healy, Amy Plimmer, Jane Salmon, Josie Smith, Drew Turner, Jana Zitha.

## Author Contributions

**Conceptualization:** Cary James, Erika Duffell.

**Data curation:** Katherine C. Sharrock, Erika Duffell.

**Formal analysis:** Katherine C. Sharrock, Erika Duffell.

**Investigation:** Erika Duffell.

**Methodology:** Katherine C. Sharrock, Erika Duffell.

**Project administration:** Katherine C. Sharrock, Erika Duffell.

**Supervision:** Erika Duffell.

**Validation:** Erika Duffell.

**Visualization:** Erika Duffell.

**Writing – original draft:** Katherine C. Sharrock.

**Writing – review & editing:** Katherine C. Sharrock, Teymur Noori, Maria Axelsson, Maria Buti, Asuncion Diaz, Olga Fursa, Greet Hendrickx, Cary James, Irena Klavs, Marko Korenjak, Mojca Maticic, Antons Mozalevskis, Lars Peters, Rafaela Rigoni, Magdalena Rosinska, Kristi Ruutel, Eberhard Schatz, Thomas Seyler, Irene Veldhuijzen, Erika Duffell.

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
