## [Decision Letter · Decision Letter 0]

3 May 2022

PGPH-D-22-00226

Monitoring progress towards elimination of hepatitis B and C in the EU/EEA

Dear Dr. Duffell,

Thank you for submitting your manuscript to PLOS Global Public Health. After careful consideration, we feel that it has merit but does not fully meet PLOS Global Public Health’s publication criteria as it currently stands. Therefore, we invite you to submit a revised version of the manuscript that addresses the points raised during the review process.

Please submit your revised manuscript by . If you will need more time than this to complete your revisions, please reply to this message or contact the journal office at globalpubhealth@plos.org. Please include the following items when submitting your revised manuscript:

We look forward to receiving your revised manuscript.

Kind regards,

Larissa Otero

Academic Editor

Journal Requirements:

State what role the funders took in the study. If the funders had no role in your study, please state: “The funders had no role in study design, data collection and analysis, decision to publish, or preparation of the manuscript.”

2. Please update your Competing Interests statement. If you have no competing interests to declare, please state: “The authors have declared that no competing interests exist.”

3. In the online submission form, you indicated that “Data can be made available upon request.”. All PLOS journals now require all data underlying the findings described in their manuscript to be freely available to other researchers, either 1. In a public repository, 2. Within the manuscript itself, or 3. Uploaded as supplementary information.

4. Please provide separate figure files in .tif or .eps format only and remove any figures embedded in your manuscript file. Please also ensure that all files are under our size limit of 10MB.

5. We notice that your supplementary tables is included in the manuscript file. Please remove them and upload them with the file type 'Supporting Information'. Please ensure that each Supporting Information file has a legend listed in the manuscript after the references list.

Additional Editor Comments (if provided):

Reviewers' comments:

Reviewer's Responses to Questions

**Comments to the Author**

1. Does this manuscript meet PLOS Global Public Health’s publication criteria? Is the manuscript technically sound, and do the data support the conclusions? The manuscript must describe methodologically and ethically rigorous research with conclusions that are appropriately drawn based on the data presented.

Reviewer #1: Yes

Reviewer #2: Partly

2. Has the statistical analysis been performed appropriately and rigorously?

Reviewer #1: N/A

Reviewer #2: N/A

3. Have the authors made all data underlying the findings in their manuscript fully available (please refer to the Data Availability Statement at the start of the manuscript PDF file)?

Reviewer #1: Yes

Reviewer #2: No

4. Is the manuscript presented in an intelligible fashion and written in standard English?

Reviewer #1: Yes

Reviewer #2: Yes

5. Review Comments to the Author

Reviewer #1: Comments to the authors

The authors present findings from a survey of EU/EEA countries regarding progress towards elimination of hepatitis B and C. Overall, despite minimal available data, the report highlights the importance of measuring key indicators to track progress at the regional and country level. Additionally, among those who reported data, it is clear that even high-income countries are falling short of elimination goals.

Major comments:

1. It is highly concerning that the sources of data were lacking for many countries. The authors fully own this as a limitation, but it calls into question the validity of the reported results. Can the authors provide examples of the “modelling or surveillance data”? Could these sources be confirmed in any way? Can the data be replicated? This needs to be addressed prior to publication.

2. Authors should provide context for global hepatitis B and C elimination efforts beyond the EU and EEA. What percentage of the 31 countries are considered high-income? What are the implications for low-income countries if high-income countries are falling short of reporting and elimination goals?

3. Why did authors choose to report based on 2020 rather than 2030 SDG goals? The report is outdated (reporting 2017 data in 2022) and I believe readers would be more interested in looking forward rather than backwards. The authors reference both 2020 and 2030 goals, making it unclear which targets are being evaluated here.

4. Can authors comment on whether the chosen indicators should be shifted? Perhaps the criteria for the goals are too stringent such that the goals will not be achievable? For instance, Polaris Observatory Collaborators recommend a shift to using absolute targets as a better measure of true elimination (https://onlinelibrary.wiley.com/doi/abs/10.1111/jvh.13412).

5. Regarding infant vaccination, the authors should report the vaccination schedules for the various countries, whether in the main file or in a supplementary file. It is unclear as written what the different schedules are and which countries include a birth dose of HBV vaccine. This makes the measurements shown in Table 2 difficult to interpret. It is striking that Denmark, Finland and Iceland do not have national policies. Perhaps the authors should recommend universal vaccination in the Discussion?

6. Consider restructuring the discussion section to focus separately on HBV and HCV rather than combining.

Minor comments:

1. Introduction – goal 3, “to promote health and wellbeing”, should be placed in quotations.

2. Introduction – last sentence of 1st paragraph – remove “in 2015” since you referenced 2017 at the beginning of the sentence. Otherwise, make it clear to what “in 2015” refers.

3. Introduction – paragraph 3, insert a comma before “in alignment with the SDGs”.

4. Materials and Methods – paragraph 1, insert a comma after “in consultation with an advisory group”

5. Results – Completeness and data sources section – 2nd sentence of 2nd paragraph – what is meant by “older treatment data”?

6. Results – Policy section – Start a new sentence beginning, “Of the 31 countries…”

7. Results – Continuum of care section – Chronic HBV and HCV cases diagnosed – 2nd and 3rd paragraphs - Recommend moving the list of the 4 and 7 countries as follows: “four (Denmark, Ireland, Netherlands, UK (Scotland)) of the 12 countries…”

8. Results – Continuum of care section – Recommend changing the title of subsection ii to “Eligible cases treated”

9. Discussion – paragraph 2 – please provide a citation and/or overview for the “data…previously collected by WHO Europe” regarding vertical HBV transmission.

10. Discussion – paragraph 4 – 2nd sentence – a word seems to be missing after “next monitoring”. Perhaps “next monitoring period”? When is this planned?

11. Discussion – paragraph 5 – 7th sentence – remove the comma after “healthcare settings”

12. Discussion – paragraph 6 – what is meant by “sampling for exampling”?

13. Discussion – paragraph 8 – recommend rewording the 1st sentence for clarity’s sake

14. Discussion – paragraph 8 – authors should provide a citation if making a claim such as “it is likely that the majority of those on treatment are virally suppressed”

15. Discussion – paragraph 12 – 1st sentence - remove the comma after “carried out in 2021”

16. Discussion – final paragraph – 4th sentence – add a comma between “monitoring systems” and “ensuring adequate resources…”

17. General – Please revise the following run-on sentences and/or sentences with unclear wording:

o Materials and Methods – 1st sentence

o Results – Completeness and data sources section – 1st sentence of 2nd paragraph

o Discussion – 1st sentence of 5th paragraph; 4th and 7th sentences of 6th paragraph

Reviewer #2: The manuscript by Sharrock, et al. attempts to quantify progress toward HBV and HCV elimination in the European region. The data is already five years old limiting its utility and the most robust portion of the study appears to be data from other reports (e.g., WHO/UNICEF, EMCDDA...), which are already publicly available. The data collected by ECDC has major gaps including well known avaiable data that are collected/reported by countries as part of their national registry. No attempt was made to collect national registry data but instead the preferred methodology was to send a questionaire to the countries to fill out on their own.

When limited data is available, researchers are faced with two options – not publish the data because they do not meet a minimum quality threshold, or, publish data and state the limitations. The latter option is influenced by a key decision – does our newly reported data add to the knowledge of the field? The authors believe that the reported data, despite numerous limitations, expand our knowledge with regards to HBV and HCV elimination.

There are some major gaps in the study. HBV birth dose vaccination rates that are part of every WHO guideline are missing and only discussed briefly in the discussion section. In addition, total infections for HCV and HBV are missing as well as the references for the original sources. The authors state that national representatives provided this data but what were the sources and the quality of this data. Finally, there are an obvious mythology issue. The percent treated of total diagnosed favors countries with low diagnosis rate. Thus, for HCV, Romania and Bulgaria rank higher than France and UK which have had well established national elimination programs. For HBV, the authors state the complexity of assessing progress for HBV due to treatment eligibility criteria. It would have been very helpful if they had collected and reported country specific HBV treatment eligibility criteria. That would have expanded our knowledge of the field.

Please allow me to disagree with the authors decision to submit this work for publication. The data is old/outdated, there are major gaps in the data, and insufficient references to original data have been provides. This study does not expand our knowledge but in fact leads to confusion.

Below are some detailed comments:

• Page 3, paragraph 2 – It is stated that “Elimination is defined as a 65% reduction in hepatitis-related deaths and a 90% reduction in new chronic HBV and HCV infections compared to the 2015 baseline”. The new guidelines by WHO use absolute targets which should be used here (World Health Organization, Interim guidance for country validation of viral hepatitis elimination, N.A.C. Geneva: World Health Organization, Editor. 2021.)

• Page 4, paragraph 4 – authors state “Table 2 provides a summary of the data provided by countries on the WHO European Region…”. The data in Table 2 came from other sources listed in the methodology section (WHO/UNICEF, EMCDDA report..). This should be stated clearly as these other sources have their own limitations which the reader can only understand after reading those reports.

• There is no mention of Table 3 in the result section.

• Table 3, page 17 – the data shown here is not consistent with available data in the country (national registries). For example, Sweden has a national registry that records both HBV and HCV diagnosed cases (over 80% diagnosed) but is listed as having not data. Germany has the RKI registry that has its limitations but does have reported diagnosed numbers for HCV and HBV. Same applies to Finland, Luxembourg, Italy…. Also, many of these countries (Poland, Slovakia, Luxembourg…) have national registries for the number of HCV and HBV treated patients that are not reported here.

• Page 4, paragraph 5 – authors state “estimates of the numbers with chronic hepatitis B and C were over five years old in around a third of countries..” but the authors made no effort to adjust for the impact of time. This is critical for HCV as publications in France have shown a continued decline in HCV prevalence as more people are treated. In addition, this publication should report the estimated number of HBV and HCV infections by country as this is used in the denominator of % diagnosed.

• Page 4, paragraph 6 – authors state “Information were obtained on the source of the data collected”. No reference for the data sources was provided.

• Page 5, paragraph 6 – authors state “and of the 31 countries, 20 (65%) reported there was a plan and 10 of these countries reported that there were funds allocated from the national budget to implement the plan.” A list of countries should be provided.

• Page 5, paragraph 5 – “The proportion of chronic HBV and HCV cases that were diagnosed as of the end of 2017 was calculated as a fraction with the number diagnosed as the numerator and the estimated number with chronic infection as the denominator”. Again, the denominator is not shown in this manuscript.

• Figure 6, page 24 – % diagnosed who are treated is the wrong metric that benefits countries who have a low diagnosis rate (Romania or Bulgaria). A country like France or UK treated many more people but they rank low because they have a high diagnosis rate.

• Figure 3, page 24 – For HCV, it is not clear if the authors are reporting a single year treated or cumulative. The WHO targets for HCV take into account cumulative HCV treated (since HCV treatment is curative). Please see the WHO report and they way the interpreted the HCV treatment rate (World Health Organization, Global progress report on HIV, viral hepatitis and sexually transmitted infections, 2021. Accountability for the global health sector strategies 2016–2021: actions for impact. 2021, World Health Organization: Geneva, Switzerland.)

• Page 15, Table 1 – it is not clear to why the goal of 90% universal coverage of HBV birth dose is not included in the core indicators for measuring progress towards the SDG targets. Every WHO guidance includes this target.

• Page 15, Table 1, Page 4, paragraph 3 – only programmatic targets are included. The authors state that Incidence and mortality data (impact targets) were not included because data were not available. However, on page 4, paragraph 1, the authors show Eurostat (population and mortality data) as a data source. What is this data used for? In discussion section, exclusion and plans for collection impact targets (mortality and incidence) should be discussed further as every WHO report of viral hepatitis targets cover both programmatic and impact targets.

6. PLOS authors have the option to publish the peer review history of their article (what does this mean?). If published, this will include your full peer review and any attached files.

**Do you want your identity to be public for this peer review?** For information about this choice, including consent withdrawal, please see our Privacy Policy.

Reviewer #1: **Yes: **Peyton Jessie Thompson, MD, MSCR

Reviewer #2: No

---

## [Editor Report · Decision Letter 1]

5 Jul 2022

Monitoring progress towards elimination of hepatitis B and C in the EU/EEA

PGPH-D-22-00226R1

Dear Dr Duffell,

We are pleased to inform you that your manuscript 'Monitoring progress towards elimination of hepatitis B and C in the EU/EEA' has been provisionally accepted for publication in PLOS Global Public Health.

Best regards,

Larissa Otero

Academic Editor
